# Sublethal Effects of Chlorantraniliprole on *Spodoptera litura* (Lepidoptera: Noctuidae) Moth: Implication for Attract-And-Kill Strategy

**DOI:** 10.3390/toxics9020020

**Published:** 2021-01-22

**Authors:** Fanfang Kong, Yaqin Song, Qian Zhang, Zhongyue Wang, Yongqiang Liu

**Affiliations:** 1State Key Laboratory for Biology of Plant Diseases and Insect Pests, Institute of Plant Protection, Chinese Academy of Agricultural Sciences, Beijing 100193, China; fanfang103@126.com (F.K.); zhangqian9405@163.com (Q.Z.); wangzhy0301@sina.com (Z.W.); 2Guangxi Academy of Specialty Crops, Guilin 541004, China; wrongpiano@163.com

**Keywords:** *Spodoptera litura*, moth, chlorantraniliprole, sublethal effects, reproduction

## Abstract

The integrated use of plant-derived volatile attractants and synthetic insecticides in attract-and-kill programs is a useful tool for integrated pest management programs reducing pesticide input. Efficient alternative insecticides are critically needed to replace methomyl, which has been banned on cruciferous vegetables in China because it is also highly toxic to nontarget organisms. In the present study, among 15 commonly used insecticides were screened for toxicity against *S. litura* moths, where chlorantraniliprole, flubendiamide, and emamectin benzoate was found to have the highest levels of toxicity (LC_50_ of 0.56, 3.85, and 6.03 mg a.i. L^−1^ respectively). After exposure to the low lethal concentration LC_50_ of chlorantraniliprole, fecundity of the moths was substantially reduced. Egg-hatching was lower for LC_20_- and LC_50_-treated moth pairs than for untreated control pairs. Net reproductive rate (*R*_0_), intrinsic rate of increase (*r*), and finite rate of increase (*λ*) were significantly reduced in LC_50_♀ × LC_50_♂ cohorts. Larval mortality was significantly higher in subsequent generations in pairs of LC_50_-treated moths. Chlorantraniliprole, which was most toxic and had significant sublethal effects on moths, can be used as an alternative insecticide to methomyl in the attracticide for controlling *S. litura* moths, and the LC_50_ indicated a high potential for efficacy in the control *S. litura* through attract-and-kill schemes.

## 1. Introduction

The tobacco caterpillar, *Spodoptera litura* Fab. (Lepidoptera, Noctuidae), is a serious pest of many agricultural crops, such as soybean (*Glycine max* L.), maize (*Zea mays* L.), and vegetables and fruit trees in the southern and eastern regions of Asia [1,2,3]. Losses from their feeding can range from 26 to 100% in the field [3]. At present, chemical insecticides are still the main method to control *S. litura*, and they usually target the larvae [4].

In China, the range of *S. litura* in China can be divided into three zones: the year-round breeding region, the overwintering region, and the summer breeding region [5,6,7]. Due to the high reproductive capacity of the adults and their ability to migrate long distances, populations of the pest can expand rapidly in the summer breeding region [8,9]. Therefore, the adult stage is also the key target for managing pest populations.

The attract-and-kill strategy is a potential tactic in the management of agricultural pests that presupposes the intelligent combination of an attracting agent (e.g., host kairomone) and a killing agent (e.g., insecticide) [10]. Such an approach can significantly decrease egg production and subsequent larval populations [11]. Recently, attractants consisting of synthetic plant volatiles (aromatic information compounds) and toxicants have been developed for trapping both sexes of noctuidae adults including *S. litura* [12]. In China, researchers have developed an attracticide for lepidopteran pests of cotton (*Gossypium* spp.), maize (*Zea mays* L.), and peanut (*Arachis hypogaea* L.) [13,14,15]. Methomyl is presently the major pesticide used for this purpose [16]; however, its use on cruciferous vegetables has been banned in China because it is highly toxic to nontarget organisms [17]. Therefore, an alternative to methomyl is needed to use in attracticides.

The extensive literature on the toxicity and effectiveness of various insecticides against *S. litura*, is almost entirely concentrated on the larvae [18,19,20] because the larvae harms plants directly and are the usual targets of insecticides. Less information is available on the oral toxicity of insecticides against adults of *S. litura*. An attracticide must rapidly incapacitate and kill moths to prevent them from laying eggs before they die; thus, the concentration of an insecticide in the attracticide is important. Because all pesticides can elicit sublethal effects on pests [21,22], we need to investigate the sublethal effects of insecticides that are highly toxic to *S. litura* moths so that the dosage can be reduced in attractants.

Against this background, here we evaluated the susceptibility of *S. litura* moths to 15 common insecticides. Further, we investigated the sublethal effects of the most effective insecticide, chlorantraniliprole, on adults of *S. litura* and highlight that chlorantraniliprole at low doses would be effective in the attract-and kill against *S. litura*.

## 2. Materials and Methods

### 2.1. Insect and Insecticide

*S. litura* moths were captured by a searchlight trap at the Langfang Experimental Station (39.53° N, 116.70° E), Chinese Academy of Agricultural Sciences (CAAS), in Hebei Province, China. Then, they were stored in cages with mesh sides for egg collection. The larvae of *S. litura* were reared on an artificial diet [23,24] under conditions of 25 ± 1 °C and 60 ± 5% relative humidity with a 14:10 light:dark photoperiod. All moths were provided with a solution of 10% *v/v* sugar and 2% *v/v* vitamin complex for nutrition supplement. Adults of the third and fourth generations were used for bioassays. For experiments on sublethal effects of chlorantraniliprole on *S. litura* moths, the insects were sexed after pupation and placed separately in ventilated plastic cages for emergence. Moths were used within 24 h of emergence. 

Insecticides were technical grade formulations (% *w*/*v*, as indicated) of 15 insecticides were tested as follows: chlorantraniliprole (95.3%), cyhalothrin (95%), thiodicarb (95%), flubendiamide (98%), abamectin (97%), spinosad (90%), indoxacarb (94%), emamectin benzoate (92%), chlorfenapyr (94.5%), beta-cypermethrin (96.5%), fenpropathrin (92%), fenvalerate (96%), chlorpyrifos (98%), carbosulfan (90%). All insecticides were provided by the Institute for the Control of Agrochemicals (ICA), the Ministry of Agriculture (MOA), China. A spinetoram suspension concentrate (SC) (50,000 mg a.i. L^−1^) was obtained from Langfang, China (Produced by Dow AgroSciences, UK).

### 2.2. Bioassay of S. litura Moths in the Laboratory

A 100 ml stock solution (10,000 mg a.i. L^−1^) of spinetoram was prepared in distilled water, while all other stock solutions (50,000 mg a.i. L^−1^) of insecticides were diluted using dimethyl sulfoxide (DMSO). Each stock solution was diluted using a 10% *v/v* honey solution containing 0.1% *v/v* Tween-80 to the desired experimental concentrations (Beijing Chemical Reagent Co. Ltd., Beijing, China). The 10% honey solution containing 1% DMSO and 0.1% Tween-80 was used as a blank control. To prevent insect contact with the insecticide solution, the cotton ball with insecticide or honey solution was placed on the bottom of a plastic cup (7.4 cm, top diameter; 9.7 cm, height). Next, a bottomless plastic cylinder that was 9.5 cm high (7.4 cm, top diameter) was wrapped with two layers of cotton gauze and placed inside this cup, maintaining ~0.8 mm between the gauze and soaked cotton ball. Five *S. litura* moths (3 days old) were randomly chosen and placed in each cylinder as one replicate; all bioassay treatments had five replications. All cylinders were maintained at 25 ± 1 °C, 60 ± 5% RH, and had a photoperiod of 14: 10 h (L: D). The mortality rate of moths in the cylinder was observed after 24 h. For sublethal effects of chlorantraniliprole to *S. litura* moths, LC_20_, LC_50_, and LC_90_ values were calculated.

### 2.3. Sublethal Effects on Reproduction

Five *S. litura* moths of the same sex were placed into a plastic cup, as described in Section 2.2. The cotton ball with chlorantraniliprole at sublethal doses (LC_20_ and LC_50_) and 10% honey solution was placed on the bottom of the plastic cup. To obtain enough live moths treated with different doses of chlorantraniliprole, 80 replicates (400 females and 400 males) were subjected to the LC_50_ treatment, 60 replicates (300 females and 300 males) to the LC_20_ treatment, and 60 replicates (300 females and 300 males) to the control treatment. 

After 24 h, the female and male moths from the different treatments were paired. The mating patterns were as follows: LC_50_♀ × LC_50_♂, LC_50_♀ × CK♂, CK♀ × LC_50_♂, LC_20_♀ × LC_20_♂, LC_20_♀ × CK♂, CK♀ × LC_20_♂, and CK♀ × CK♂. Each pair was placed in a plastic cup (7.4 cm, top diameter; 9.7 cm, height), and 10% honey solution was provided as food for moths. The top of the plastic cup was covered with gauze for oviposition. Eggs on the gauze were counted, and new gauze was placed on the plastic daily. Egg-hatching was recorded for a random subset of approximately 150–200 eggs. For this purpose, sections of cotton gauze with *S. litura* eggs were placed in Petri dishes (3.5 cm, diameter; 1 cm, height) and examined for seven consecutive days. Adult longevity was also recorded. One hundred replicates were used for each treatment. 

### 2.4. Sublethal Effects on Traits of Offspring

The LC_20_ and LC_50_ concentrations were used to assess the sublethal effects of chlorantraniliprole on *S. litura*. To determine the LC_20_ and LC_50_ concentrations, we first generated a concentration–mortality regression line, then calculated the respective concentrations from the regression lines (see Section 3). Newly hatched *S. litura* larvae were randomly sampled from each treatment (see Section 2.3) and placed into a 24-well plate (one larvae per well) with an artificial diet. Three replicates (20 larvae per replicate) were used for each treatment. All plates were maintained in controlled incubators (27 ± 1 °C, 50 ± 10% RH, 14L: 10D). All insects were examined daily, and the developmental period, pupa mass, adult emergence (number of pupae that eclosed to adults divided by number of pupae, multiplied by 100), and larvae mortality (number of larvae that not pupated divided by number of larvae examined, multiplied by 100) were recorded.

### 2.5. Data Analysis

The median lethal concentrations, 95% confidence limits (CLs), and slope ± SE were calculated using probit analysis. One-way analysis of variance (ANOVA) was used to analyze the effects of lethal concentrations of chlorantraniliprole on female longevity, male longevity, fecundity, egg-hatching rate, developmental period, pupa weight, adult emergence, and larvae mortality, followed by Tukey’s honestly significant difference (HSD) test (*p* < 0.05) using SPSS 13.0 software (SPSS Inc., Chicago, IL, USA) [25]. The selected *S. litura* population parameters, including the net reproductive rate (R_0_), intrinsic rate of increase (r), finite rate of increase (λ), and mean generation time (T) were analyzed according to the age-stage, two-sex life table theory, using the program TWOSEX-MSChart [26]. Means, standard errors, and significant differences were calculated using a bootstrap procedure in TWOSEX-MSChart, with 100,000 replications [27]. Before analysis, all data were tested for normality and homogeneity of variances. Statistical analysis was performed using GraphPad Prism software [28].

## 3. Results

### 3.1. Insecticide Toxicity to S. litura Moths

Toxicities of insecticides to the *S. litura* moths varied considerably, but mortality was consistently <5% in the control groups (Table 1). The order of toxicity (from high to low) for the 15 insecticides was chlorantraniliprole > flubendiamide > emamectin benzoate, fenpropathrin > chlorpyrifos, fenvalerate, indoxacarb > lambda-cyhalothrin, beta cypermethrin, thiodicarb > avermectin, spinetoram, spinosad, carbosulfan, and chlorfenapyr (LC_50_ values with overlapping 95% confidence intervals were classified as having the same level of toxicity). The toxicity of chlorantraniliprole was the highest among the tested insecticides, with an LC_50_ value of 0.56 mg a.i. liter^−1^, while avermectin, spinetoram, spinosad, carbosulfan, and chlorfenapyr had the lowest toxicity, with LC_50_ values > 100 mg a.i. liter^−1^ (Table 1).

### 3.2. Lethal Effects of Chlorantraniliprole on S. litura Moths

Based on the mortality records for the six experimental treatments, the LC_20_ and LC_50_ value was 0.245 and 0.561 mg·L^−1^, respectively (Figure 1). The LC_50_ and LC_20_ value was used as the lethal and low-lethal concentrations, respectively, in subsequent experiments.

### 3.3. Adult Reproduction

The longevity of female adults was significantly reduced by LC_50_ and LC_20_ compared with the control (F = 12.46, df = 2, 18, *p* < 0.001). The longevity of male adults was reduced by the LC_50_ treatment, but not significantly compared with the control (F = 4.34, df = 2, 18, *p* = 0.029). Fecundity of female adults significantly decreased in the LC_50_♀ × LC_50_♂ cohorts compared with the control (F = 6.40, df = 6, 14, *p* = 0.002). Furthermore, the egg-hatching rate in the LC_50_♀ × LC_50_♂, LC_50_♀ × CK♂, CK♀ × LC50♂, and LC_20_♀ × LC_20_♂ cohorts was significantly lower than in the control treatments (F = 13.42, df = 6, 14, *p* < 0.001) (Figure 2).

### 3.4. F_1_ Generation Developmental Duration

For *S. litura* offspring, adult emergence for the LC_50_♀ × LC_50_♂ cohorts (65.50 ± 1.17%) was significantly lower than control treatments (85.09 ± 0.78%) (F = 4.922, df = 6, 14, *p* = 0.007). Larval mortality for the LC_50_♀ × LC_50_♂ (61.33 ± 1.76%) and CK♀ × LC50♂ cohorts (44.67 ± 4.06%) was significantly higher than for the control treatment (24.00 ± 2.00%) (F = 16.237, df = 6, 14, *p* < 0.001) (Table 2).

### 3.5. Life History Parameters

Net reproductive rate (*R_0_*) was significantly reduced for LC_50_♀ × LC_50_♂ (2.44 ± 0.58) and LC_50_♀ × CK♂ (47.99 ± 17.23) cohorts compared to the control (137.43 ± 22.43). The intrinsic rate of increase (*r*) was the lowest for LC_50_♀ × LC_50_♂ cohorts (0.03 ± 0.01), followed by the LC_50_♀ × CK♂ (0.14 ± 0.02), CK♀ × LC_50_♂ (0.14 ± 0.01), LC_20_♀ × LC_20_♂ (0.15 ± 0.01), and CK♀ × LC_20_♂ cohorts (0.15 ± 0.01), which were all significantly lower than that of the control (0.18 ± 0.01). The finite rate of increase (λ) was the lowest for LC_50_♀ × LC_50_♂ cohorts (1.03 ± 0.01), followed by LC_50_♀ × CK♂ (1.15 ± 0.02), CK♀ × LC_50_♂ (1.15 ± 0.01), LC_20_♀ × LC_20_♂ (1.16 ± 0.01), and CK♀ × LC_20_♂ cohorts (1.16 ± 0.01), which were all significantly lower than that of the control (1.20 ± 0.01) (Table 3). 

## 4. Discussion

An attracticide, a combination of synthetic plant volatiles and an insecticide, is used to trap and control lepidopteran moths [17,29,30]. However, research has focused on the toxicity and sublethal effects of insecticides on lepidopteran larvae [18,21,31,32,33,34]. There are many kinds of insecticides to be selected for larval control in most cases, and the sublethal doses of insecticides on larvae could adversely affect the developmental and reproductive traits, and lead to a population decrease [21,32,33,34]. Little work has yet explored the toxicity and sublethal effects of insecticides on *S. litura* moths. In this study of the toxicity of 15 insecticides commonly used against *S. litura* moths, chlorantraniliprole had highest toxicity against *S. litura* moths. Therefore, chlorantraniliprole is a good candidate for replacing the hazardous methomyl in attracticide products. Chlorantraniliprole at LC_20_ or LC_50_ concentrations had sublethal effects, reducing egg-hatching and the longevity of female adults, and the LC_50_ concentration reduced fecundity. These negative effects may be related to the mechanism of action of chlorantraniliprole, which acts on the ryanodine receptors in insects and affects calcium homeostasis in the cell, leading to feeding cessation, lethargy, muscle paralysis, and ultimately, death of the insect [35].

The toxicity of chlorantraniliprole, flubendiamide, and emamectin benzoate was high against the moths, in accordance with their high toxicity against larvae [3,20]. However, the moths appeared to be less susceptible to chlorfenapyr compared with their larvae [36], indicating that the developmental stages have inherent differences in their susceptibility [37]. *S. litura* larvae having the ability to develop high resistance to chlorantraniliprole [38], further research is needed on whether moths can easily develop resistance to chlorantraniliprole. The rapid incapacitation and killing of moths is critical to an effective attractant to reduce adult opportunity to lay eggs before death. The LT_50_ of chlorantraniliprole against *S. litura* moths was found to be low; therefore, its high insecticidal toxicity and rapid efficiency against target pests make it a good candidate for controlling *S. litura* moths.

In the current study with *S. litura*, the LC_50_ of chlorantraniliprole significantly reduced the fecundity of the LC_50_♀ × LC_50_♂ cohorts, as found for larvae of *Platynota idaeusalis* and *Helicoverpa zea* that fed on a diet containing a sublethal concentration of tebufenozide [39,40]. Some insecticides can decrease egg-hatching [41,42]; however, our research showed that egg-hatching was also reduced when female and male moths that had been exposed to LC_20_ and LC_50_ concentration of chlorantraniliprole were paired with untreated male and female moths. Whether the reduction in hatching rate was caused by a decrease in mating rate or the lack of hatching of fertilized eggs needs further study. Our results indicate that future use of this compound in attract-and-kill schemes may not be limited by *S. litura* adult abundance or immigration rates, because egg-hatching will drop sharply even when untreated immigrants mate with the resident, pesticide-tainted individuals. Fieldwork, however, is essential to validate these hypotheses and fine-tune pesticide delivery methods.

The intrinsic rate of increase (*r*_m_), a measure of the ability of a population to increase exponentially in an unlimited environment, provides an effective summary of an insect’s life history traits [43], which, combined with a toxicity assessment, can provide a more accurate estimate of the population-level effect of a toxic compound [44,45,46]. In our study, *r*_m_ was lower for LC_50_♀ × LC_50_♂, LC_50_♀ × CK♂ (0.14 ± 0.02), CK♀ × LC_50_♂ (0.14 ± 0.01), LC_20_♀ × LC_20_♂ (0.15 ± 0.01), and CK♀ × LC_20_♂ cohorts after treatment with chlorantraniliprole compared with the controls, meaning a population increase would be delayed.

In conclusion, the novel insecticide chlorantraniliprole had the highest toxicity and fastest activity among the insecticides tested against *S. litura* moths. Thus, a low concentration will reduce *S. litura* fecundity and egg-hatching, and slow population growth. Hence, the inclusion of a low concentration of chlorantraniliprole in an attract-and-kill delivery scheme constitutes a highly desirable alternative to broad field-level applications or coating seeds with insecticide. Such an “attract-and-kill” measure will potentially enhance environmentally friendly pest management, and thus the IPM toolbox for controlling this global agricultural pest.

## Figures and Tables

**Figure 1 toxics-09-00020-f001:**
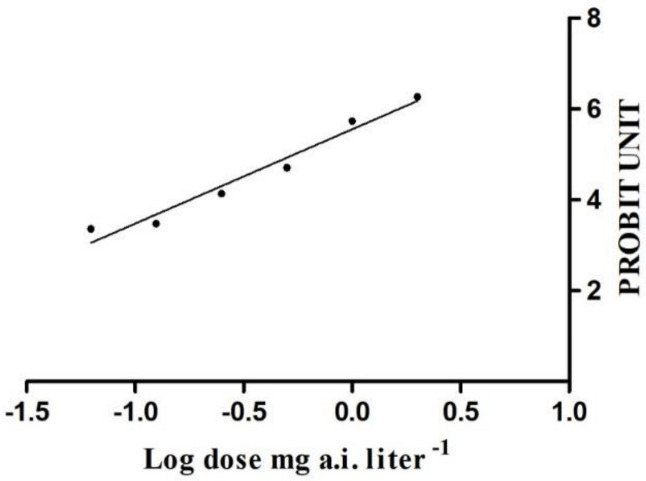
Linear regression of morality (probit unit) of *Spodoptera litura* and chlorantraniliprole concentration (logtransformed).

**Figure 2 toxics-09-00020-f002:**
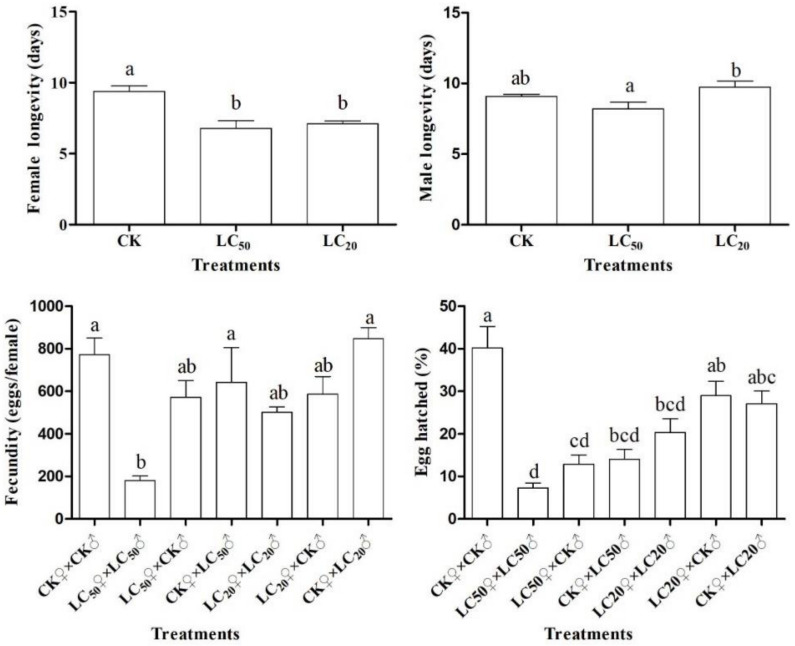
Effects of chlorantraniliprole at sublethal doses on the longevity and fecundity of *Spodoptera litura* after treatment in the adults. CK♀, LC_20_♀, and LC_50_♀ mean the surviving females from control, and LC_20_ and LC_50_ are the treated cohorts that were used to pair with males, respectively; CK♂, LC_20_♂, and LC_50_♂ mean the surviving males from control, and LC_20_ and LC_50_ are the treated cohorts that were used to pair with females, respectively. “Fecundity” is the number of eggs laid per female. Different letters above bars indicate a significant intermonth difference at the 5% level in Tukey’s HSD tests.

**Table 1 toxics-09-00020-t001:** Toxicity of 15 insecticides to *S. litura* moths (24 h).

Insecticide	Slope ± SE	LC_50_ (mg a.i. Liter^−1^)	95% Fiducial Limits	*R*^2^ (df)	*p*
Chlorantraniliprole	2.126 ± 0.170	0.56	0.48 ~ 0.67	20.506 (16)	0.1983
Flubendiamide	2.392 ± 0.237	3.85	3.25 ~ 4.53	6.314 (13)	0.934
Emamectin benzoate	1.962 ± 0.179	6.03	5.01 ~ 7.23	7.535 (16)	0.9615
Fenpropathrin	2.153 ± 0.226	7.31	6.12 ~ 8.88	4.187 (13)	0.989
Chlorpyrifos	1.814 ± 0.170	13.29	10.88 ~ 16.09	4.106 (16)	0.9987
Fenvalerate	1.714 ± 0.206	16.57	13.02 ~ 20.47	5.275 (13)	0.9686
Indoxacarb	2.224 ± 0.231	17.36	14.42 ~ 20.63	8.300 (13)	0.8234
Lambda-cyhalothrin	1.691 ± 0.163	28.12	22.94 ~ 34.48	7.432 (16)	0.964
Beta cypermethrin	1.681 ± 0.162	33.83	23.34 ~ 51.08	44.476 (16)	0.0002
Thiodicarb	2.362 ± 0.235	41.16	34.72 ~ 48.54	5.284 (13)	0.9684
Avermectin		>100			
Spinetoram		>100			
Spinosad		>100			
Carbosulfan		>100			
Chlorfenapyr		>100			

**Table 2 toxics-09-00020-t002:** Effects of chlorantraniliprole at sublethal doses on the life history traits of the offspring of *Spodoptera litura* after treatment in the adults.

Treatment	Development								
	1st Instar (Days)	2nd Instar (Days)	3rd Instar (Days)	4th Instar (Days)	5th Instar (Days)	6th Instar (Days)	Pupa Weight (mg)	Adult Emergence (%)	Larval Mortality (%)
CK♀ × CK♂	3.36 ± 0.07 d	2.69 ± 0.08 d	2.95 ± 0.10 b	3.00 ± 0.11 bc	3.12 ± 0.12 cd	3.86 ± 0.15 a	452.86 ± 10.58 ab	85.09 ± 0.78 a	24.00 ± 2.00 cd
LC_50_♀ × LC_50_♂	3.46 ± 0.14 cd	2.99 ± 0.09 cd	3.00 ± 0.16 b	3.37 ± 0.07 ab	3.14 ± 0.18 c	4.07 ± 0.18 a	411.90 ± 16.41 b	65.50 ± 1.17 b	61.33 ± 1.76 a
LC_50_♀ × CK♂	5.14 ± 0.16 a	3.14 ± 0.01 bc	3.04 ± 0.06 b	2.49 ± 0.05 c	2.18 ± 0.02 d	1.91 ± 0.10 c	503.88 ± 14.90 a	88.14 ± 3.77 a	38.67 ± 4.06 bc
CK♀ × LC_50_♂	3.52 ± 0.15 bcd	3.39 ± 0.11 ab	4.72 ± 0.44 a	3.67 ± 0.27 a	3.14 ± 0.16 c	2.76 ± 0.18 b	418.02 ± 12.37 b	77.51 ± 2.92 ab	44.67 ± 4.06 ab
LC_20_♀ × LC_20_♂	4.09 ± 0.13 b	3.42 ± 0.08 ab	2.40 ± 0.06 b	3.23 ± 0.07 ab	4.84 ± 0.22 b	2.15 ± 0.01 c	413.47 ± 18.77 b	73.11 ± 8.63 ab	15.33 ± 8.35 d
LC_20_♀ × CK♂	3.84 ± 0.06 bcd	3.48 ± 0.03 ab	2.28 ± 0.02 b	3.22 ± 0.01 ab	5.27 ± 0.22 b	1.99 ± 0.01 c	397.45 ± 4.47 b	82.56 ± 1.17 ab	23.33 ± 2.40 cd
CK♀ × LC_20_♂	3.99 ± 0.20 bc	3.57 ± 0.10 a	2.53 ± 0.06 b	3.40 ± 0.09 ab	6.49 ± 0.28 a	1.98 ± 0.05 c	421.74 ± 5.48 b	88.79 ± 1.91 a	16.00 ± 3.06 d

The abbreviations CK♀, LC_20_♀, and LC_50_♀ indicate, respectively, the surviving females from the control and LC_20_- and LC_50_-treated cohorts that were paired with males; CK♂, LC_20_♂, and LC_50_♂ indicate, respectively, the surviving males from the control and LC_20_- and LC_50_-treated cohorts that were paired with females. Different letters within a column indicate a significant intermonth difference at the 5% level in Tukey’s HSD tests.

**Table 3 toxics-09-00020-t003:** Effects of chlorantraniliprole at sublethal doses on the life-table parameters of *Spodoptera litura* after treatment in the adults.

Treatments	Net Reproductive Rate (*R*_0_)	Intrinsic Rate of Increase (*r*)	Finite Rate of Increase (*λ*)	Mean Generation Time (*T*)
CK♀ × CK♂	137.43 ± 22.43 a	0.18 ± 0.01 a	1.20 ± 0.01 a	27.35 ± 0.54 d
LC_50_♀ × LC_50_♂	2.44 ± 0.58 c	0.03 ± 0.01 c	1.03 ± 0.01 c	28.72 ± 0.47 cd
LC_50_♀ × CK♂	47.99 ± 17.23 b	0.14 ± 0.02 b	1.15 ± 0.02 b	27.99 ± 0.54 d
CK♀ × LC_50_♂	88.68 ± 29.43 ab	0.14 ± 0.01 b	1.15 ± 0.01 b	31.39 ± 0.99 ab
LC_20_♀ × LC_20_♂	110.58 ± 31.51 ab	0.15 ± 0.01 b	1.16 ± 0.01 b	31.02 ± 0.58 ab
LC_20_♀ × CK♂	123.39 ± 28.81 a	0.16 ± 0.01 ab	1.17 ± 0.01 ab	30.02 ± 0.50 bc
CK♀ × LC_20_♂	140.63 ± 29.46 a	0.15 ± 0.01 b	1.16 ± 0.01 b	32.47 ± 0.57 a

CK♀, LC_20_♀, and LC_50_♀ mean the surviving females from control, and LC_20_ and LC_50_ are the treated cohorts used to pair with males, respectively; CK♂, LC_20_♂, and LC_50_♂ mean the surviving males from control, and LC_20_ and LC_50_ are the treated cohorts used to pair with females, respectively. Values that are followed by a different letter(s) within a column differed significantly at *p* < 0.05 using the bootstrap procedure in the TWOSEXMS Chart, with 100,000 replications, for *R*_0_, *r*, *λ*, and *T*).

## Data Availability

The data presented in this study are available on request from the corresponding author. The data are not publicly available due to privacy.

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
