# Peer review of "Sublethal Effects of Chlorantraniliprole on Spodoptera litura (Lepidoptera: Noctuidae) Moth: Implication for Attract-And-Kill Strategy"

_toxics, 2021, doi:10.3390/toxics9020020_

Round 1

Reviewer 1 Report

The aim of this paper was to evaluate the potential of insecticide to be use in the control of a major pest Spodoptera litura in a combination with attractant. The authors have also evaluated the effects of the insecticide on the offspring and their major life history traits. The manuscript is of interest and the experiments are well conducted. A few points can be improved. Major points - In Introduction, lines 61-62 “highlight the efficacy of chlorantraniliprole at low doses when combined with feeding attractants against the moths in the field.” This sentence should be modified because the authors did not show that. All the experiments have been done only with the insecticide, there is not mention of the attractants or field. Please correct that. - In Materials and Methods, section on “sublethal effects on traits of offspring”, the authors should give more details to explain how is calculated adult emergence % and larval mortality %. This is not very clear for me at the reading of the results of Table 2, the first line is coherent with 85% for adult emergence and 24% for larval mortality but how can the authors have 65% of adult emergence with 61% larval mortality? Adult emergence is not calculated from the number of larvae present at the beginning of the experiment. - Several times in the manuscript, the authors address aspects related to “integrated pest management programs” however they never consider an important point which is insecticide resistance. Resistance of S. litura to chlorantraniliprole has already been detected in China and the use of this insecticide could be compromised. This important point should at least be noted in the discussion. - Minor points: - Spodoptera litura should be in italic in the title (line 3) and in the keywords (line 28)

Author Response

  • In Introduction, lines 61-62 “highlight the efficacy of chlorantraniliprole at low doses when combined with feeding attractants against the moths in the field.” This sentence should be modified because the authors did not show that. All the experiments have been done only with the insecticide, there is not mention of the attractants or field. Please correct that.

Response: Thank you for the comments and we have revised it.

  • In Materials and Methods, section on “sublethal effects on traits of offspring”, the authors should give more details to explain how is calculated adult emergence % and larval mortality %. This is not very clear for me at the reading of the results of Table 2, the first line is coherent with 85% for adult emergence and 24% for larval mortality but how can the authors have 65% of adult emergence with 61% larval mortality? Adult emergence is not calculated from the number of larvae present at the beginning of the experiment.

Response: Thank you for the comments and we have revised it in line 119-121.

  • Several times in the manuscript, the authors address aspects related to “integrated pest management programs” however they never consider an important point which is insecticide resistance. Resistance of litura to chlorantraniliprole has already been detected in China and the use of this insecticide could be compromised. This important point should at least be noted in the discussion.

Response: Thank you for the comments and we have revised it on line 215-217.

  • Spodoptera litura should be in italic in the title (line 3) and in the keywords (line 28)

Response: Thank you for the comments and we have revised it.

Reviewer 2 Report

Dear Authors, dear Editor,

draft article “Sublethal effects of chlorantraniliprole … toxics-1042483-peer-review-v1” reports on a standard toxicological study aimed at selecting the most active larvicide to be paired with a chemo-attractant to formulate a product to fight  the moths of the tobacco caterpillar, Spodoptera litura. The Authors employ standard techniques in this field of applicative research and describe their findings in a systematic and clear way. Their work can be interesting for practitioners in the field who may find their article easily in the Toxics journal. This draft can be easily revised to publication by amending or improving a few points.

In Table 1 the LC50 concentration units are not reported (it is mg a.i./L).

In the envisioned attract-and-kill strategy this article only reports the selection of chlorantraniliprole as the “kill” agent, without dealing with the attractant and the overall formulation. Interested readers may be curious to understand more on the attractant (a bait or a species-specific chemoattractant?) and the overall strategy, if such information is available.

I have a methodological issue on mating experiments: the groups are CKm, CKf, LC20m, LC20f, LC50m, LC50f. Complete combinatorial experiments lead to 6x6=36 experiments, while only 7 are reported. Many interested, but unspecialized readers would ask the same queston, so a comment would be helpful to strenghten the value of the Authors’ work.

I would recommend publication after this brief revision.

Best regards

Author Response

  • In Table 1 the LC50 concentration units are not reported (it is mg a.i./L).

Response: Thank you for the comments and we have revised it.

  • In the envisioned attract-and-kill strategy this article only reports the selection of chlorantraniliprole as the “kill” agent, without dealing with the attractant and the overall formulation. Interested readers may be curious to understand more on the attractant (a bait or a species-specific chemoattractant?) and the overall strategy, if such information is available.

Response: Thank you for the comments and the bioattract consisting of synthetic plant volatiles (Aromatic information compounds) and toxicants, and it was added on line 44-45.

  • I have a methodological issue on mating experiments: the groups are CKm, CKf, LC20m, LC20f, LC50m, LC50f. Complete combinatorial experiments lead to 6x6=36 experiments, while only 7 are reported. Many interested, but unspecialized readers would ask the same queston, so a comment would be helpful to strenghten the value of the Authors’ work.

Response: Thank you for the comments and in the design of the experiment, we designed two levels and control. We only considered the situation of different levels and control within different levels, but did not consider the situation between different levels. In the future experiments, we will seriously consider the situation between different levels.
